



# Validation of Sentinel-1 offshore winds and average wind power estimation around Ireland

Louis de Montera[1], Tiny Remmers[1], Cian Desmond[1], Ross O'Connell[1]

[1]MaREI Centre for Marine and Renewable Energy, Beaufort Building, Environmental Research Institute, University College
5   Cork, Ringaskiddy, Ireland

*Correspondence to*: Cian Desmond (cian.desmond@ucc.ie)

**Abstract.** In this paper, surface wind speed and average wind power derived from Sentinel-1 Synthetic Aperture Radar Level
OCN product were validated against four weather buoys and three coastal weather stations around Ireland. A total of 1,544
match-up points was obtained over a two-year period running from May 2017 to May 2019. The match-up comparison showed
that the satellite underestimated the wind speed compared to in situ devices, with an average bias of 0.4 m/s, which decreased
linearly as a function of wind speed. Long-term statistics using all the available data, while assuming a Weibull law for the
wind speed, were also produced and resulted in a significant reduction of the bias. Additionally, the average wind power was
found to be consistent with in situ data, resulting in an error of 10% and 5% for weather buoys and coastal stations, respectively.
These results showed that the Sentinel-1 Level 2 OCN product can be used to estimate the wind speed distribution, even in
coastal areas. Maps of the average and seasonal wind speed and wind power illustrated that the error was spatially dependent,
which should be taken into considerations when working with Sentinel-1 SAR data.

## 1 Introduction

With the ever-increasing interest in offshore wind energy and the rise of floating turbines, the estimation of the available wind
energy over large offshore areas has become necessary. According to the Global Wind Energy Council (Global wind statistics
2014, http://www.gwec.net/wp-content/uploads/2015/02/GWEC_GlobalWindStats2014_FINAL_10.2.2015.pdf), offshore
wind power costs are expected to reduce by about 45% by 2050. One factor that can be associated with cost reduction is the
increasing availability of accurate remote sensing data over large areas with a high resolution which can significantly reduce
project risk at site finding stage. Moreover, the measurement of offshore wind speed contributes to the understanding of marine
phenomena and boundary layer processes. Low altitude meteorological parameters, such as wind, are therefore key parameters
in the modelling of the Earth system.

Several studies have already attempted to assess the offshore wind energy potential using spaceborne scatterometers, such as
ERS-1, ERS-2, NSCAT, QuickSCAT and ASCAT (Sánchez et al., 2007; Pimenta et al., 2008; Karagali et al., 2014; Bentamy
and Croize-Fillon, 2014; Remmers et al., 2019). However, the spatial resolution of these instruments is at best 12.5 km², which
prevents the assessment in coastal areas (0-20 km from the shore) and the study of fine sub-mesoscale processes that can affect



turbine yields and climate processes. In this framework, spaceborne Synthetic Aperture Radar (SAR) sensors offer a much higher spatial resolution, allowing for wind speed retrieval with a level of detail not discernible from scatterometer data.

In this study, the Sentinel-1 A and B Level 2 OCN product produced by the European Space Agency (ESA) was validated. This SAR instrument records neutral surface winds at 10 m above sea level (a.s.l) with a spatial resolution of 1 $km^2$. Even though this type of analysis was previously performed in other parts of Europe (Hasager et al., 2015), it has never been

conducted in Ireland, which has a significant offshore wind resource (Remmers et al., 2019). Additionally, to the authors' knowledge, the Sentinel-1 level 2 OCN product has not yet been validated against in situ measurements, with the exceptions of one match-up comparison in the waters adjacent to the Korean peninsula (Jang et al., 2019) and another focusing on comparison with coastal lidar (Ahsbahs et al., 2017). Similarly, long term statistics retrieved using this product, such as the average wind power, which is the most relevant for the wind energy industry, have never been analysed before.

Sentinel-1 A and B satellites were launched by ESA with C-band SARs on April 3rd 2014 and April 22nd 2016 respectively. These satellites have been fully operational around the coast of Ireland since May 2017. Sentinel-1 SAR sensors have a swath width of 250 km and an incidence angle ranging from 29.1° to 46°. Surface winds measured by Sentinel-1 A and B were compared with four offshore buoys in situ data and three coastal weather stations around Ireland. The wind distributions were also estimated based on the empirical histogram and the assumption that they followed a Weibull law, and compared to the

measured in situ distributions. The effects of the low temporal sampling of the satellite on the long-term statistics were assessed. Finally, a map of the average wind speed and wind power available around Ireland and its seasonal variations was presented with a spatial resolution of 1 $km^2$.

## 2 Data and Methodology

### 2.1 Sentinel-1 SAR Level 2 OCN

Sentinel-1 A and B are two polar-orbiting satellites equipped with C-band SAR. This sensor has the advantage of operating at wavelengths not impeded by cloud cover or a lack of illumination and can acquire data over a site during day or night in all weather conditions. The Sentinel-1 Level 2 OCN product includes a component called Ocean Wind Fields (OWI) which is a ground range gridded estimate of the surface wind speed and direction at 10 m a.s.l, assuming a neutral atmospheric

stratification, with a spatial resolution of 1 $km^2$. The two satellites are located on the same orbit 180° apart and at an altitude close to 700 km. In Irish coastal waters, the acquisition mode is Interferometric Wide (IW) swath using the TOPSAR technique. All Sentinel-1A and B SAR images in IW acquisition mode from May 1, 2017 to May 1, 2019, in the area located around Ireland between 51°N and 56°N in latitude and 5°W and 16°W in longitude, were collected (n=5,509). The quality flag for these data ranges from 0 to 3 (0 being the best and 3 the worst) and, following visual inspection, only data with a quality flag

≤ 2 were used for the validation. The Level 2 product tiles were aggregated into a gridded map for the area of interest, in order



to form a data cube where each pixel had a corresponding time series of measurements. The revisit rate is ranges from 10 to 20 passes per month for most areas in Irish waters, which occur in the morning around 6.30 am or in the evening around 6 pm. Figure 1 shows the number of samples for each pixel and Figure 2 shows the average daily passing time of the satellites.


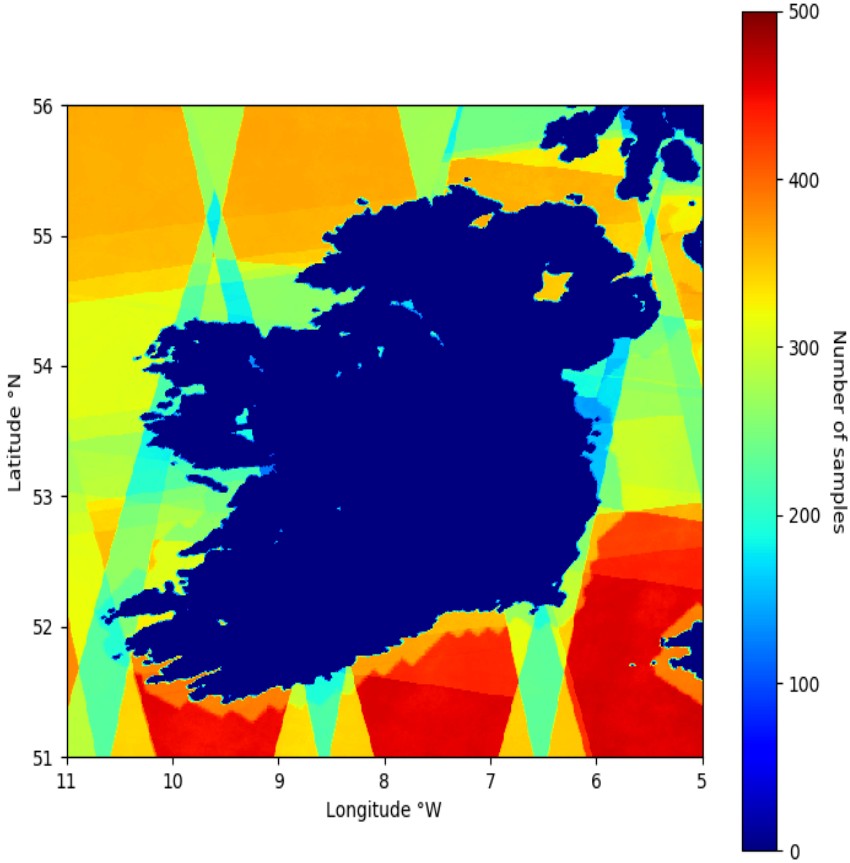

**Figure 1: Number of Sentinel-1 A and B passes across Ireland over a two-year period running from May 2017 to May 2019 with an acceptable quality flag (≤ 2).**

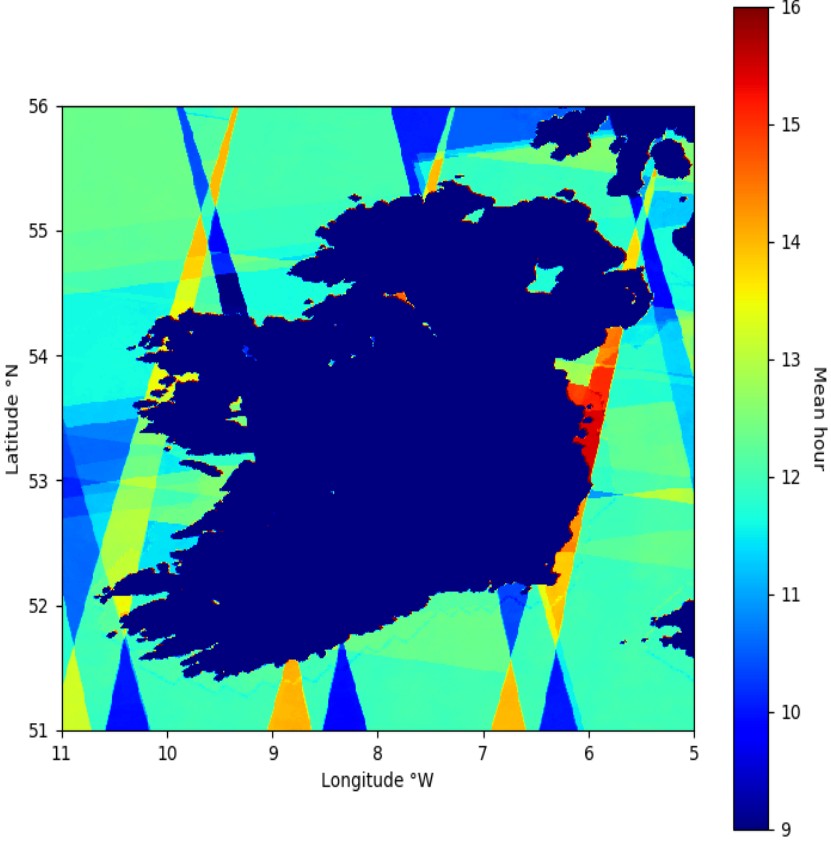


**Figure 2: Average daily hour of Sentinel-1 A and B passes across Ireland over a two-year period running from May 2017 to May 2019 with an acceptable quality flag (≤ 2).**

## 2.2 In situ instruments

### 2.2.1 Weather Buoys

Ireland's Marine Institute operates five offshore weather buoys named M2, M3, M4, M5 and M6. Their location is shown on Figure 3. The data from these were downloaded from the Marine Institute website with a two-year time series ranging from May 1st 2017 to May 1st 2019. The hourly product corresponds to the wind speed averaged over a period of 10 mn every hour at 3 m a.s.l.. As a result of extensive maintenance periods, the buoys are not always functioning leading to a lack of

measurements in the dataset, up to several months, for some locations. Due to this phenomenon, and to a poor offshore coverage frequency from Sentinel-1 satellites, the M6 buoy was excluded in the validation analysis.

In order to compare Sentinel-1 SAR Level 2 OCN product with this network of instruments, the in situ buoy measurements
were extrapolated from 3 m to 10 m a.s.l.. The following log law was used, assuming a neutral atmospheric stratification
(Carvalho et al., 2017):

$$U_{10} = \frac{\ln\left(\frac{Z_{sat}}{Z_0}\right)}{\ln\left(\frac{Z_{buoy}}{Z_0}\right)} \cdot U_{buoy} \tag{1}$$

where $U_{10}$ is the wind speed at 10 m in m s$^{-1}$, $U_{buoy}$, the wind speed measured by the buoys in m s$^{-1}$, $Z_{sat}$ the altitude of the
satellite measurements in m, $Z_{buoy}$ the altitude of the buoy measurements in m, and $Z_0$ the roughness length of the sea surface
taken as 0.0002 m (Barthelmie et al., 2005). Table 1 gives the exact locations of these buoys and their percentage of availability.

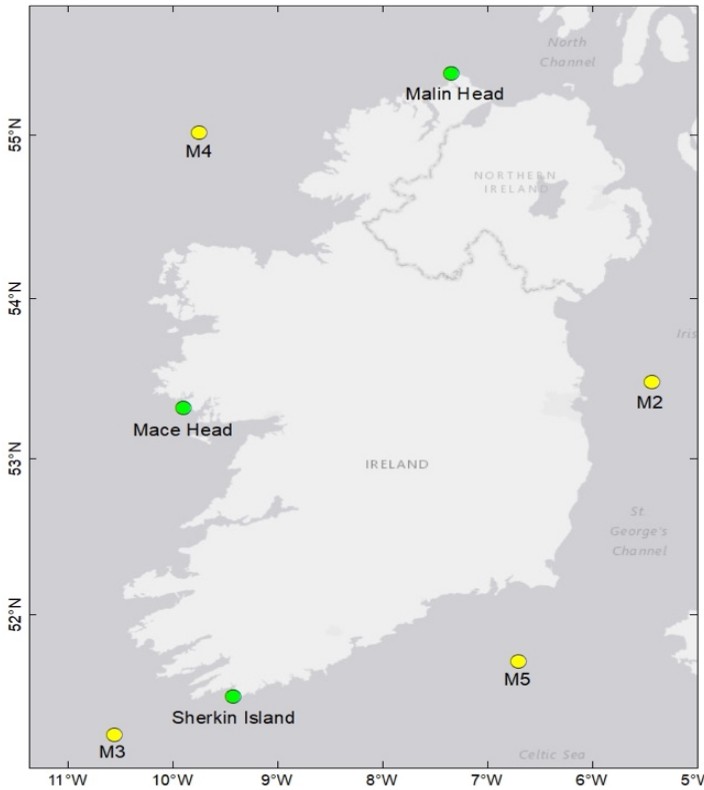

**Figure 3: Location of metocean buoys (yellow) and coastal weather stations (green) used in the validation of Sentinel-1 SAR surface
winds.**



| Name | Type | Latitude | longitude | Altitude in m | % of availability |
|------|------|----------|-----------|---------------|-------------------|
| M2 | Metocean buoy | 53.48°N | 05.42°W | 3 | 63 |
| M3 | Metocean buoy | 51.21°N | 10.55°W | 3 | 59 |
| M4 | Metocean buoy | 55.00°N | 09.99°W | 3 | 72 |
| M5 | Metocean buoy | 51.69°N | 06.70°W | 3 | 85 |


**Table 1: Location and characteristics of the weather buoys used in the comparison with Sentinel-1 SAR Level 2 OCN product.**

### 2.2.2 Coastal weather stations

Three weather stations operated and maintained by Met Éireann, the Irish weather forecasting service, were used to validate

the Sentinel-1 SAR Level 2 OCN wind speeds in coastal areas. These three stations were considered for the validation analysis because they are located close to the shore (less than 200 m, see Figure 3), at a low altitude (approx. 20 m), and far from any hills or relief. The stations are situated on the west coast of Ireland at Sherkin Island, Mace Head, and Malin Head, and have continuous wind speed records during the two-year period of study (Table 2). The most probable situation by far for the west coast of Ireland is that the wind is flowing from the sea towards the land. Simulations of these type of flows have shown that

for a moderate coastal slope, onshore wind speeds recorded at proximity to the shore can equate the wind speeds at sea just before reaching the coast (Bassi Marinho Pires et al., 2015). Therefore, the wind speed derived from satellite measurements were not scaled to the altitude of the weather stations, but instead they were considered as being on the same streamline. The weather station data were compared with Sentinel-1 SAR Level 2 OCN wind speeds measured 1 or 2 km away from the shore in order to avoid land contamination.


| Name | Type | Latitude | Longitude | Altitude in m | % of availability |
|------|------|----------|-----------|---------------|-------------------|
| Sherkin Island | Weather station | 51.47°N | 9.42°W | 21 | 100 |



| Mace Head | Weather station | 53.32°N | 9.90°W | 21 | 100 |
| Malin Head | Weather station | 55.37°N | 7.34°W | 20 | 100 |

**Table 2: Location and characteristics of the coastal weather stations used in the comparison with Sentinel-1 SAR Level 2 OCN product.**


## 2.3 Assessment criteria

The error $e_i$ between Sentinel-1 Level 2 OCN wind speed, denoted $U_i$, and the in situ measurement, denoted $u_i$, is defined as follows:

$e_i = U_i - u_i$ (2).

The criteria used in the comparison were the mean error (or bias), the standard deviation ($\sigma$), the Root Mean Square Error (RMSE), the Mean Absolute Error (MAE) and the linear correlation coefficient (R), respectively defined by:


$Bias = \frac{1}{N}\sum_{i=1}^{N} e_i$ (3)


$\sigma = \sqrt{\frac{1}{N-1}\sum_{i=1}^{N}(e_i - Bias)^2}$ (4)

$RMSE = \sqrt{\frac{1}{N}\sum_{i=1}^{N} e_i{}^2}$ (5)

$MAE = \frac{1}{N}\sum_{i=1}^{N}|e_i|$ (6)



$\quad R = \frac{1}{\sigma_U \sigma_u (N-1)} \sum_{i=1}^{N} (U_i - U)(u_i - u)$ (7)

where $U$ and $u$ denote the mean of satellite and in situ wind speeds respectively, $\sigma_U$ and $\sigma_u$ their standard deviation, and $N$ the number of match up samples.

**2.4 Wind distribution estimation**

The average wind power density $P$ in W m$^{-2}$, simply called wind power in the following, is the average kinetic energy passing through a unit of surface per unit of time. It can be estimated directly from the wind speed time series using the following formula:

$P = 0.5\rho(1/N) \sum_{i=1}^{N} U_i^3$ (8)


where $\rho$ is the air density (1.245 g m$^{-3}$ at 10°C) and $U_i$ the wind speed. However, in order to compensate for the low number of samples provided by the satellites, some prior knowledge on the surface wind speed distribution can be used. It is assumed here that it follows a classical Weibull law which is fitted to the empirical histogram. The Weibull law probability density

function is given by:

$pdf(U) = \frac{k}{\lambda} \left(\frac{U}{\lambda}\right)^{k-1} e^{-(U/\lambda)^k}$ (9)

where $\lambda$ is a scaling parameter in m s$^{-1}$ and $k$ a dimensionless shape parameter. The parameters of the best Weibull law

corresponding to the dataset are obtained by the method of the moments (Pavia and O'Brien, 1986):

$k = (\sigma/\mu)^{-1.086}$ (10)

$\lambda = \frac{\mu}{\Gamma\left(\frac{1}{k}+1\right)}$ (11)

where $\mu$ is the mean wind speed and $\sigma$ its standard deviation. This method allows for prediction of the correct wind speed distribution without having the full information about it, thus enhancing the amount of information that can be obtained from the satellite data. In order to verify the accuracy of the method and of the satellite measurements, the parameters obtained with



this method were compared with the parameters obtained with the in situ data in the same way. The wind power as a function
of these parameters is given by the following formula (Justus et al., 1976):

$$P = 0.5 \, \rho \lambda^3 \Gamma(1 + 3/k) \tag{12}$$

where $\Gamma$ is the Gamma function.

## 3. Results

### 3.1 Match-up comparison

The main objective of the Sentinel-1 SAR surface wind comparison with in situ data was to highlight the agreement and
dissonance between the two. Sentinel-1 SAR Level 2 OCN surface wind data and in situ wind data were collocated in space
and time. Since the spatial resolution of Sentinel-1 SAR data is very high (1 km$^2$) and offshore winds have a low spatial
heterogeneity caused by sea surface homogeneity, the spatial resolution was slightly degraded in order to increase the number
of samples. A 3 km$^2$ pixel around the in situ instruments was used and the best value in this pixel for each satellite pass in
terms of quality and distance was then chosen.

In the time domain, each in situ measurement with the corresponding satellite measurement performed in a 30 mn time interval
before or after were selected for the analysis. For all buoys, the wind speed correlation at a one-hour time interval was around
0.99, which showed that the time difference between the satellite and in situ data does not introduce a significant source of
error. Another factor in this respect is that Sentinel-1 SAR spatial averaging at the resolution of 1 km$^2$ may somewhat
compensate for the lack of time averaging. However, the bias due to these differences in the measurement technique, in space
and time, is difficult to predict theoretically. Therefore, the bias can be caused not only by the SAR sensor intrinsic error, but
also by the different scales of measurement. Another source of potential error derived from the assumption of neutral
atmospheric stability when scaling the buoy data from 3 m to 10 m a.s.l using Equation (1). Hence, the overall bias needed to
be evaluated empirically through a match-up comparison.
The bias for all available data was found to be -0.42 m s$^{-1}$ and -0.39 m s$^{-1}$ and the RMSE 1.41 m s$^{-1}$ and 1.51 m s$^{-1}$ for the buoys
and weather stations, respectively (Table 3 & 4). These results showed that Sentinel-1 SAR Level 2 OCN is underestimating
the in situ wind speed. A very high linear correlation coefficient of 0.93 for the buoys and 0.92 for the weather stations
demonstrated that Sentinel-1 SAR data are suitable for estimating the local wind speed. For all locations, the number of match-
up samples over the two-year period of study was above 150, which is known to be the minimum number of samples needed
to obtain correct wind speed statistics (Bentami and Croize-Fillon, 2014). The results also showed that the errors calculated
with offshore buoys or coastal stations are very consistent. Therefore, it can be concluded that, taking the bias into account,



Sentinel-1 SAR can be used to estimate the wind speed up to 1 km from the shore, which is the resolution of the instrument and the required distance to avoid land contamination.

| Buoy | N samples (SAR) | Mean (SAR) | Mean (in situ) | Bias | Percentile 90% (SAR) | Percentile 90% (in situ) | RMSE | MAE | R |
|---|---|---|---|---|---|---|---|---|---|
| M2 | 179 | 8.29 | 8.58 | -0.29 | 13.73 | 13.64 | 1.41 | 1.12 | 0.94 |
| M3 | 161 | 7.86 | 8.31 | -0.45 | 13.31 | 13.10 | 1.74 | 1.12 | 0.89 |
| M4 | 219 | 8.86 | 9.00 | -0.14 | 13.98 | 14.25 | 1.35 | 1.01 | 0.94 |
| M5 | 242 | 7.6 | 8.34 | -0.74 | 13.08 | 13.39 | 1.14 | 0.81 | 0.95 |
| Total | 801 | 8.15 | 8.57 | -0.42 | 13.52 | 13.59 | 1.41 | 1.02 | 0.93 |

**Table 3: Results of the match-up comparison of satellite measured wind speeds in m s⁻¹ with in situ measured wind speeds from weather buoys.**

| Buoy | N samples (SAR) | Mean (SAR) | Mean (in situ) | Bias | Percentile 90% (SAR) | Percentile 90% (in situ) | RMSE | MAE | R |
|---|---|---|---|---|---|---|---|---|---|
| Sherkin Island | 297 | 6.15 | 6.17 | -0.12 | 10.86 | 10.80 | 1.47 | 1.15 | 0.92 |
| Mace Head | 206 | 7.61 | 8.36 | -0.75 | 12.66 | 13.63 | 1.42 | 1.11 | 0.94 |
| Malin Head | 240 | 7.91 | 8.34 | -0.43 | 13.37 | 13.89 | 1.55 | 1.23 | 0.92 |
| Total | 743 | 7.12 | 7.52 | -0.39 | 12.30 | 12.77 | 1.51 | 1.18 | 0.93 |

**Table 4: Results of the match-up comparison of satellite measured wind speeds in m s⁻¹ with in situ measured wind speeds from coastal weather stations.**





The bias was found to be wind speed dependent. Figure 4 (left) shows that the bias was stronger at small wind speed values and reduced as the wind speed increased. This is consistent with the fact that Sentinel-1 SAR uses the sea state in order to

estimate surface winds. Indeed, low wind speeds do not necessarily cause a significant effect on the sea state and, consequently, the instrument does not always accurately estimate the surface winds. This problem is already well known and often leads to an unrealistically high number of very low wind speed values. This can be seen on the scatter plot in Figure 4 (right), which also confirmed the results related to the bias.

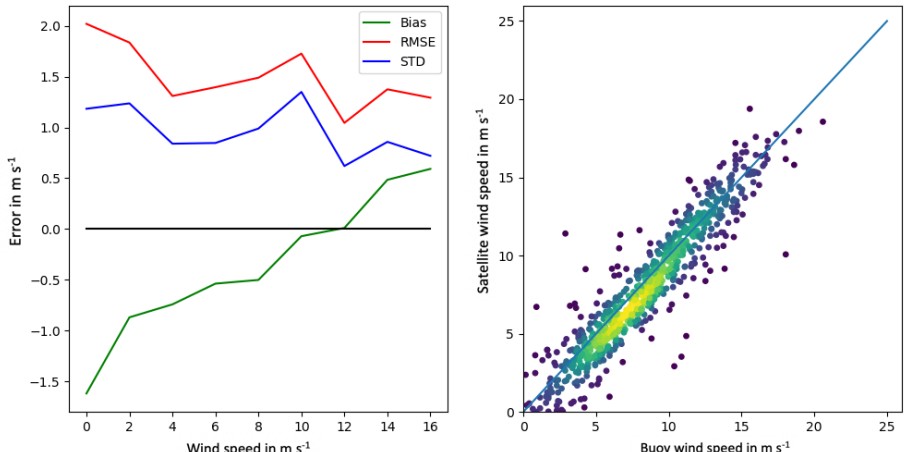


**Figure 4: Statistical representation of the Sentinel-1 Level 2 OCN error against weather buoy data as a function of SAR wind speeds (left), and scatter plot versus weather buoy data (right).**


As expected, the satellites also underestimated the wind power. The average error in the wind power was 6% for the weather buoys and 13% for the coastal weather stations, respectively (Tables 5 & 6). Since the wind power is proportional to the cube of the wind speed, a higher error (approx. 20%) would be expected. However, since the underestimation is mainly affecting low wind speed values and not so much strong values, the resulting error on the wind power was reduced. The higher bias for

two of the coastal weather stations, namely, Mace Head and Malin head, may be caused by generally lower wind speeds near the coast and, therefore, the effect of the bias was amplified at those locations.





| Buoy | K (SAR) | K (in situ) | λ (SAR) | λ (in situ) | Wind power in W m⁻² (SAR) | Wind power in W m⁻² (in situ) | % of error on wind power |
|---|---|---|---|---|---|---|---|
| M2 | 2.19 | 2.34 | 9.37 | 9.68 | 613 | 641 | -4.28 |
| M3 | 2.18 | 2.44 | 8.87 | 9.37 | 524 | 564 | -7.04 |
| M4 | 2.41 | 2.56 | 9.99 | 10.14 | 689 | 693 | -0.47 |
| M5 | 2.12 | 2.51 | 8.58 | 9.40 | 485 | 559 | -13.19 |
| Total | 2.22 | 2.46 | 9.20 | 9.65 | 578 | 614 | -6.24 |

**Table 5: Comparison of wind speed long-term statistics obtained from the four weather buoys with the ones obtained from the SAR data.**

| Buoy | K (SAR) | K (in situ) | λ (SAR) | λ (in situ) | Wind power in W m⁻² (SAR) | Wind power in W m⁻² (in situ) | % of error on wind power |
|---|---|---|---|---|---|---|---|
| Sherkin Island | 1.75 | 1.86 | 6.91 | 7.06 | 315 | 311 | 1.48 |
| Mace Head | 2.12 | 2.19 | 8.59 | 9.44 | 487 | 627 | -22.41 |
| Malin Head | 2.40 | 2.28 | 8.92 | 9.41 | 492 | 601 | -18.07 |
| Total | 2.09 | 2.11 | 8.14 | 8.64 | 431 | 513 | -13.00 |

**Table 6: Comparison of wind speed long-term statistics obtained from the three coastal weather stations with the ones obtained from the SAR data.**






### 3.2 Impact of intra-diurnal variability

The main limitation of satellite remote sensing to accurately assess the offshore wind resource derives from their reduced temporal coverage and revisit time at a given location. Since wind speeds can have strong daily variations, the impact due to the lack of intra-diurnal measurements needs to be investigated. To do so, for each match-up between the satellites and the in

situ instruments, all the in situ measurements from that 24 h period were added to the in situ data before computing the statistics (Table 7). The bias and the error on the wind power assessment were increased on average by approx. 10%. It can be concluded that the lack of intra-diurnal satellite data has a relatively small impact on the results. Since the satellites pass different locations at different times of day, some in situ locations were more affected than others. However, the increase of error on the wind power due to intra-diurnal variability was always below 7% of the total wind power.


| Buoy | Bias in m s$^{-1}$ | Bias in m s$^{-1}$ (including in situ intra-day data) | % of error on wind power | % of error on wind power (including in situ intra-day data) |
|---|---|---|---|---|
| M2 | -0.29 | -0.48 | -4.28 | -11.10 |
| M3 | -0.45 | -0.68 | -7.04 | -14.36 |
| M4 | -0.14 | -0.2 | -0.47 | -2.50 |
| M5 | -0.74 | -0.84 | -13.19 | -15.32 |
| Sherkin Island | -0.12 | -0.32 | 1.48 | -6.04 |
| Mace Head | -0.75 | -0.78 | -22.41 | -25.28 |
| Malin Head | -0.43 | -0.21 | -18.07 | -13.11 |
| Total | -0.42 | -0.50 | -9.14 | -10.82 |

**Table 7: Increase in the bias and the error on the wind power when intra-diurnal data of in situ measurements are taken into account, compared with the same results obtained for the match-up comparison.**



### 3.3 Impact of the scarce temporal coverage

In this section all the available in situ data over the two-year period of study were taken into account, including days for which there was no satellite pass. In order to compare statistics derived from the same time periods, the histograms of in situ data were computed using all of the available periods and the histogram of satellite data with satellite measurements available during these periods (Figures 5 & 6). These figures showed that, although the histograms produced from the satellite data exhibited important discrepancies compared to the one produced from the in situ data, the SAR measurements were nonetheless

sufficient to correctly estimate the Weibull laws describing wind speed statistics (in red for Sentinel-1 Level 2 OCN and in green for in situ devices in the figures). The analysis revealed a strong overall agreement between the in situ and SAR wind speed distributions, as can be seen in Tables 8 & 9. The Weibull parameters and the corresponding wind powers had very similar results, with wind power errors below approx. 10% and approx. 5% for the weather buoys and the coastal weather stations, respectively. These results were quite remarkable given the fact that the wind power is proportional to the cube of the

wind speed, meaning that its calculation has a strong magnifying effect on the error. This also means that Sentinel-1 SAR is able to retrieve the average wind power over large areas with a high spatial resolution and a reasonable error.

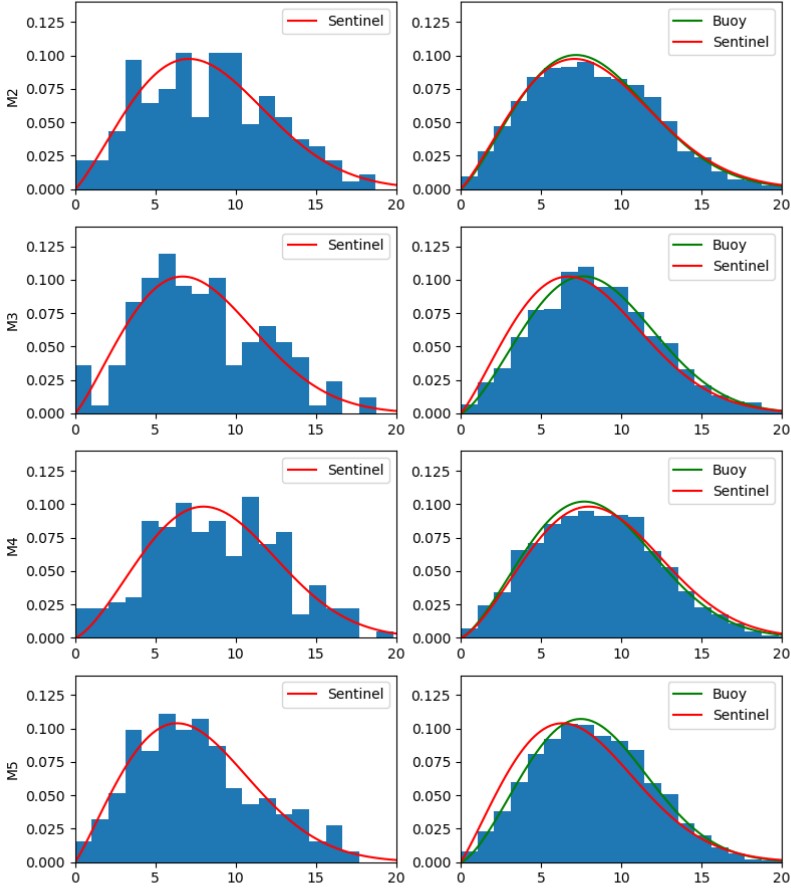

**Figure 5: Wind speed histograms in m s⁻¹ with corresponding Weibull fits for the weather buoy data compared with those produced from the SAR data at the same locations.**

| Buoy | K (SAR) | K (in situ) | λ (SAR) | λ (in situ) | Wind power in W/m2 (SAR) | Wind power in W/m2 (in situ) | % of error on wind power |
|------|---------|-------------|---------|-------------|--------------------------|------------------------------|--------------------------|
| M2 | 2.19 | 2.26 | 9.37 | 9.31 | 613 | 586 | 4.69 |
| M3 | 2.18 | 2.41 | 8.87 | 9.56 | 524 | 604 | -13.22 |
| M4 | 2.41 | 2.41 | 9.99 | 9.62 | 689 | 615 | 11.99 |
| M5 | 2.12 | 2.45 | 8.58 | 9.27 | 485 | 544 | -10.93 |





**Table 8: Comparison of the long-term wind speed statistics produced from the weather buoy data with those produced from the SAR data at the same locations.**

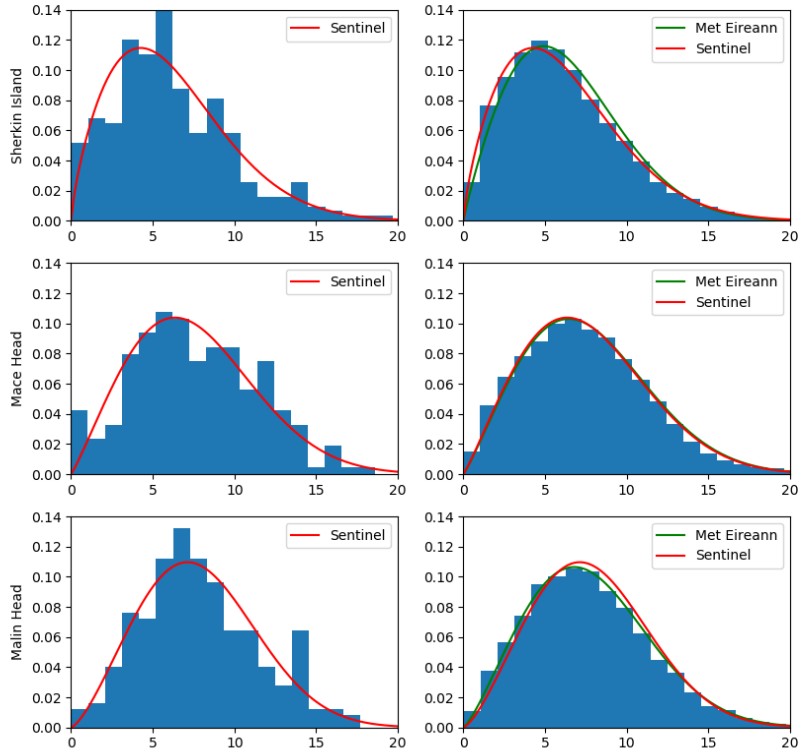


**Figure 6. Wind speed histograms in m s$^{-1}$ with corresponding Weibull fits for the coastal weather station data compared with those produced from the SAR data at the same locations.**

| Buoy | K (SAR) | K (in situ) | λ (SAR) | λ (in situ) | SAR Wind power (W/m²) | In situ Wind power (W/m²) | % of error on wind power |
|---|---|---|---|---|---|---|---|
| Sherkin Island | 1.75 | 1.92 | 6.91 | 7.21 | 315 | 319 | -1.08 |
| Mace Head | 2.12 | 2.13 | 8.59 | 8.69 | 487 | 502 | -2.99 |
| Malin Head | 2.40 | 2.26 | 8.92 | 8.78 | 492 | 492 | 0.15 |




**Table 9: Comparison of the long-term wind speed statistics produced from the coastal weather station data with those**
**produced from the SAR data.**

It is particularly interesting that the percentage error on the average wind power was lowest for the coastal weather stations. This may indicate that they could be more reliable than weather buoys, perhaps due to the presence of waves and the relatively
low altitude of the buoys. In that case, the error in offshore locations could be overestimated due to inaccuracies with the weather buoy data, although there is no possibility of proving this with certitude.

Another interesting feature is that the bias observed in the match-up comparison seemed to disappear in this climatological analysis. The main difference between the match-up comparison and the analysis performed here arises from including in situ data even when satellite data were not available. In this study, satellite data can be unavailable for two reasons: no data were
recorded as a consequence of the relatively low revisit time of the satellite, or the data recorded were discarded if it was flagged as 'bad quality'. The former should not have any effect on the long-term statistics. However, the latter might actually introduce an artificial bias in the match-up comparison by limiting it to a specific type of situation in which satellite measurements are easier to perform. For example, if good quality flags are more likely to correspond to turbulent situations, then the different scales at which the measurements are performed (10 minutes for in situ devices and 1 km$^2$ for the satellite) can introduce a
discrepancy. In that case, measurements in space will be less affected by the turbulence and closer to the average long-term distribution due to Kolmogorov's laws (Kolmogorov, 1941) stipulating that the variability linked to turbulence scales as function of $\Delta t^{1/2}$ in time and only as a function of $\Delta x^{1/3}$ in space. Finally, when the in situ database includes all types of situations, the in situ distributions converge towards the one obtained with the satellite data.

**3.4 Wind resource assessment in Irish coastal waters**

In this section, the use of the Sentinel-1 Level 2 OCN product to assess wind resources around Ireland at 10 m a.s.l. with a 1 km$^2$ spatial resolution is discussed. A clear separation of the mean wind speed into two different areas was clearly visible (Figure 7). The northwest area, starting above 53 °N and going until the beginning of the North Channel between Ireland and Scotland, was characterised by a climate of strong winds (above 9 m s$^{-1}$), while the rest of the map had a more moderate wind climate, with a mean generally around 8 m s$^{-1}$. This was consistent with the observations obtained from spaceborne
scatterometers (Remmers et al., 2019).

In terms of wind power, the results logically revealed a similar pattern with an increased heterogeneity, due to the fact that the wind power is connected to the cube of the wind speed (Figure 8). The northwest area had an average wind power of 700 W m$^{-2}$ in comparison with 500 W m$^{-2}$ for the rest of the map, resulting in an overall difference of 20% between the two areas. It is interesting to note that the central area of the Irish sea also has a significant potential in terms of wind power, although lower
than that of the northwest area. Regarding coastal areas, a steep horizontal gradient was observed from the shore up to 15-20

km offshore, with the exception of the remote peninsulas on the west coast where the gradient was much shorter or non-existent.

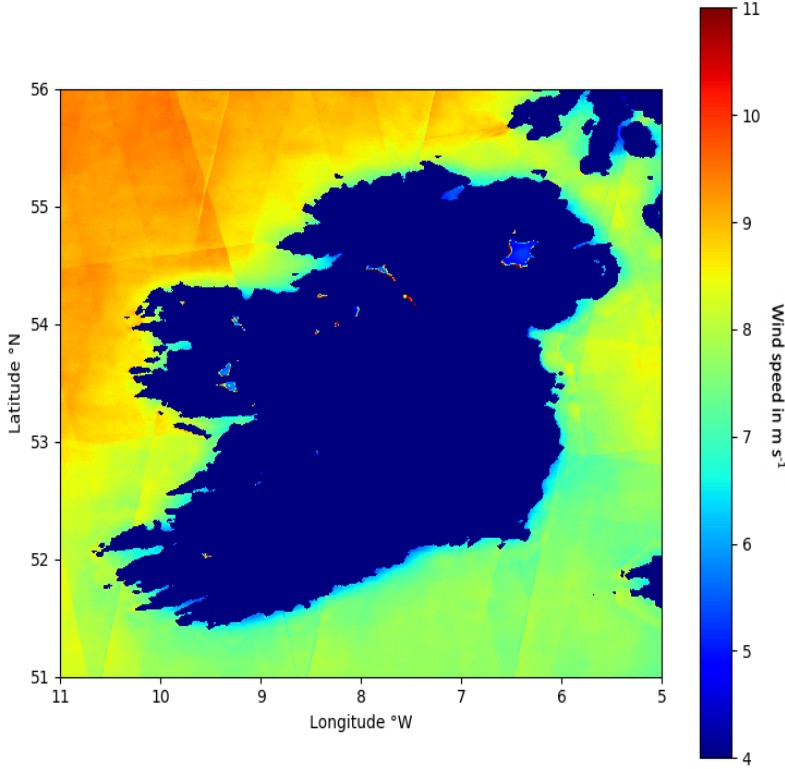

**Figure 7: Average wind speed off Ireland over a two-year period running from May 2017 to May 2019 retrieved using the Sentinel-**
**1 SAR Level 2 OCN product.**



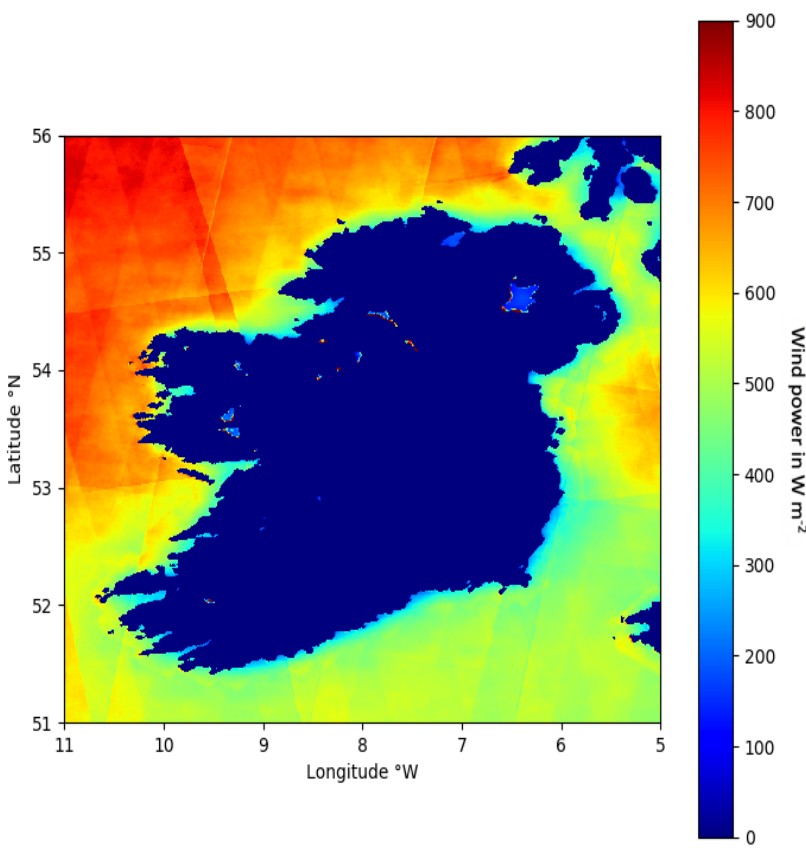

**Figure 8: Wind power off Ireland over a two-year period running from May 2017 to May 2019 retrieved using the Sentinel-1 SAR Level 2 OCN product.**


The seasonal averages of wind speed and wind power showed expected trends of low and strong winds typical of the summer and winter seasons, respectively (Figures 9 & 10). Autumn was also associated with strong winds, which corolated to the

cyclonic activity in the North Atlantic Ocean ending their trajectory in this area of Western Europe. The wind climate during spring was much more moderate than that of autumn. However, it is more turbulent and its direction more diverse, leading to a near disapearrance of the horizontal gradient in coastal areas.

As shown in Figures 7 to 10, the tracks of the satellites were still visible. This discrepancy can be related to several factors such as instrument bias associated with the incidence angle, difference in the number of samples (Figure 1) affecting the quality





of the Weibull fits, or simply a difference in the average time of the day at which the satellites pass (Figure 2) resulting to a different impact of the intra-diurnal variability. Unfortunately, no clear correlation was found between these factors and the anomalies on the maps. It was only found that the edges of the swaths have more unrealistic values, which could be due to the incidence angle or the instrument thermal noise. As a consequence, a margin of 5 pixels (roughly equivalent to 5 km) was removed from the swaths before creating the maps. The areas with less observations also had a less reliable assessment of the

mean wind speed and power, however, this limitation should disappear in the future as more samples will become available. It can be concluded that the accuracy was dependent upon location, which is a factor that should be considered when using Santinel-1 SAR data. The results also highlighted the necessity for additional in situ validation points for satellite products and showed that there is a need to improve the Sentinel-1 level 2 OCN product algorithm, perhaps through the application of machine learning techniques.


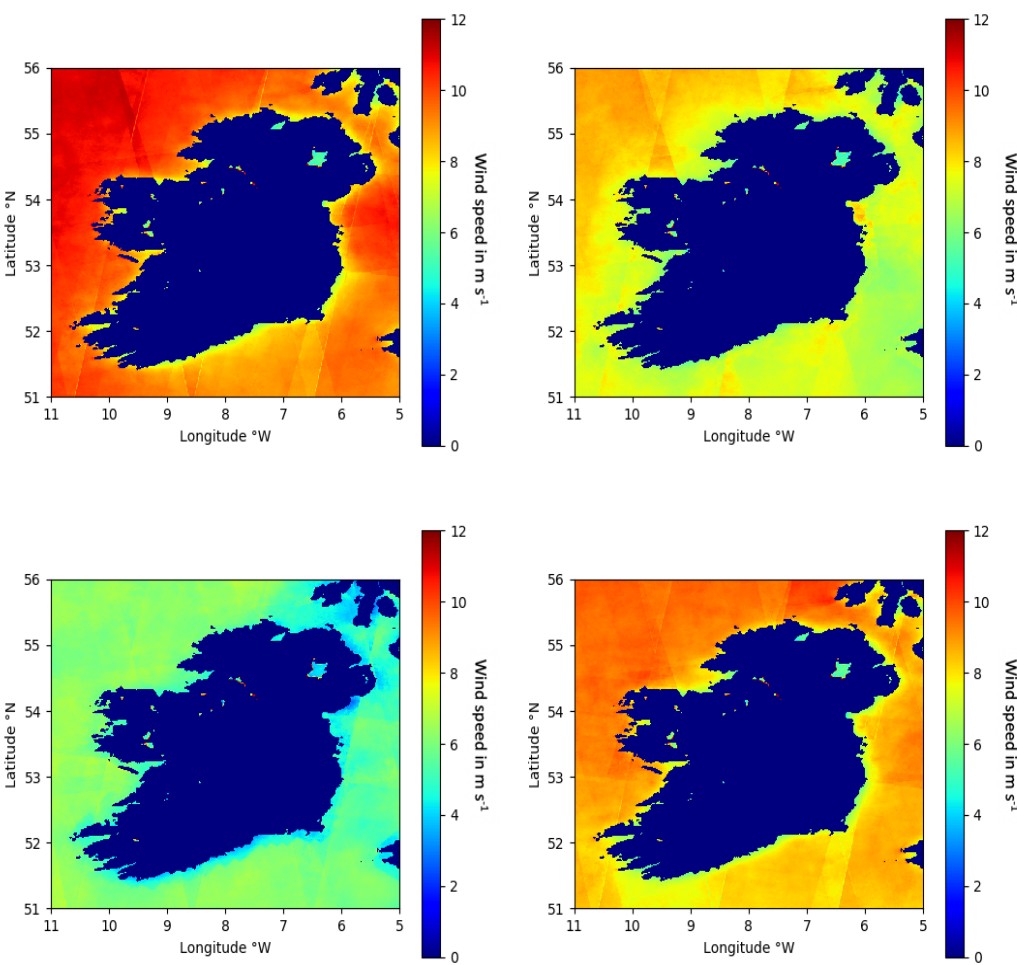

**Figure 9: Seasonal average wind speed off Ireland over a two-year period running from May 2017 to May 2019 retrieved using the Sentinel-1 SAR Level 2 OCN product (top left: winter, top right: spring, lower left: summer, lower right: autumn).**




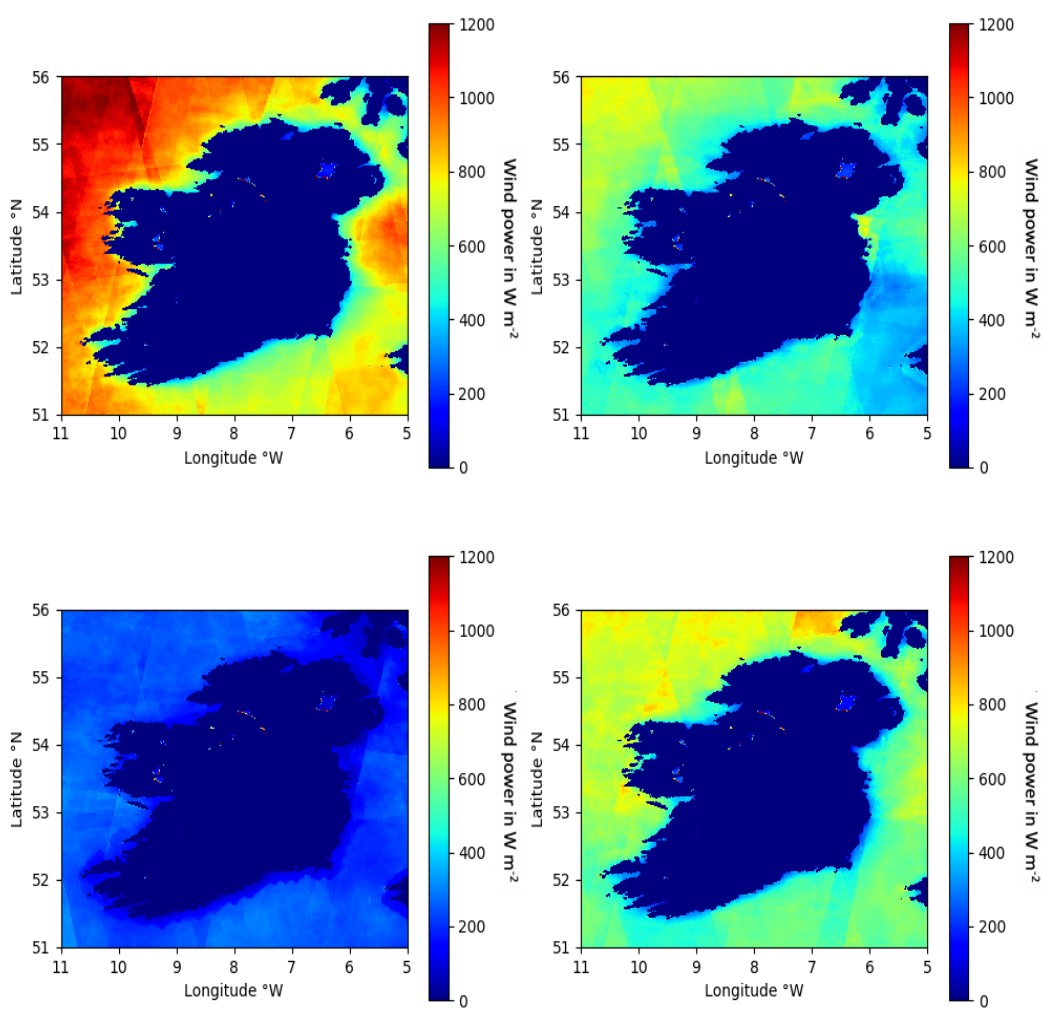

**Figure 10: Seasonal wind power off Ireland over a two-year period running from May 2017 to May 2019 retrieved using the Sentinel-1 SAR Level 2 OCN product (top left: winter, top right: spring, lower left: summer, lower right: autumn).**



## 4. Conclusions


Measurements from the Sentinel-1 Level 2 OCN product were compared with measurements from four weather buoys and three coastal weather stations located around Ireland. The match-up comparison indicated that the satellites underestimated the in situ data by 0.4 m s$^{-1}$ on average, with an RMSE of 1.45 m s$^{-1}$. These results were consistent between the weather buoys and the coastal weather station data. The bias was found to be stronger for low wind speeds, and to linearly decrease with an


increase of wind speed strength. However, this discrepancy disappeared when the long-term statistics were computed including all available in situ data. This could be associated with the in situ measurements performed at a very different spatial scale to that of the satellite measurements (a few square centimetres versus 1 km$^2$). In any case, it was concluded that the Sentinel-1 Level 2 OCN product can be used to estimate the long-term wind speed distribution and the average wind power, even in coastal areas as close as 1 km to the shore. This result could be obtained by using the method of the moments and assuming a


Weibull law in order to compensate for the low temporal coverage of the satellites.

The fact that the satellites always pass at the same hour of the day, limiting their ability to record the intra-diurnal variability, was investigated and its effects on the long-term statistics was found to be minor. Finally, the error on the average wind power was found to be on the order of 10% and 5% for weather buoys and coastal weather stations, respectively. This result was quite remarkable given the fact that the wind power is proportional to the cube of the wind speed, which strongly enhances the


original error from the wind speed. Maps of the average wind speed and wind power around Ireland were presented with a resolution of 1 km$^2$. These maps indicated that the algorithm used to develop the Sentinel-1 Level 2 OCN product needs to be improved since the satellite swaths were still visible. Users should exercise caution when working with Sentinel-1 SAR data since a location-dependent error was found. The cause of this discrepancy could not be identified, but perhaps a machine learning technique based on a learning dataset of in situ data could be used to mitigate this effect.


Future studies could focus on the combined use of SAR and scatterometer measured wind speed in order to create climatologies constructed using a longer period than the two-year period of this study. This could be particularly interesting to more accurately estimate the offshore wind energy resource. Another important application in the future would be to modify the acquisition mode in coastal areas for the satellites carrying SAR, in order to obtain the required information to estimate the wave heights. This information, only available in open seas with Sentinel-1, would be useful to correlate the wind and wave


energy and thus provide a more detailed description of the marine environment for optimising offshore wind farm siting.

## Acknowledgments

The authors would like to thank the Marine Institute for providing the offshore weather buoy data, Met Éireann for the coastal weather station data and ESA for the Sentinel-1 SAR Level 2 products.

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
