# Peer review of "Validation of Sentinel-1 offshore winds and average wind power estimation around Ireland"

_Wind Energy Science, 2019_

## Referee Comment (RC1) · Anonymous Referee #1 · 21 Oct 2019

The manuscript entitled "Validation of Sentinel-1 offshore winds and average wind power estimation around Ireland" presents a comparison between Sentinel 1 level 2 OCN and in-situ measurements. Wind field comparisons were presented between satellite product and buoys. Basically, the manuscript is a technical report presenting basic statistical results such as RMSE, correlation, MAE, among others. The manuscript must be rejected because I cannot find any scientific approach or a novel contribution. The linear comparison between satellite and in-situ data using basic statistic is not enough merit for a scientific journal.

---

## Author Comment (AC1) · 21 Oct 2019

Thank you for your comments.

The scientific value of this paper lies in the fact that it provides validation of the Sentinel-1 wind speed data against in-situ measurements. We believe that this is the first public publication to provide this validation. We have also contacted Ifremer and CLS and they have confirmed that they have validated Sentinel-1 data only against numerical models and so no "private" validation against in-situ data exists.

We feel that validation against in-situ measurements is an important step towards the

increased use of Sentinel-1 data for wind resource assessment purposes and that such validation results and details of the methodologies used should be in the public domain.

---

## Referee Comment (RC2) · Anonymous Referee #2 · 28 Oct 2019

[referee-annotated manuscript omitted]

---

## Author Comment (AC2) · 11 Dec 2019

**Comments 1 to 5**

Text edited as below to address these comments:

[revised manuscript text omitted]

*The bias for all available data used in the match-up comparison was found to be -0.42 m s-1 and -0.39 m s-1 and the RMSE 1.41 m s-1 and 1.51 m s-1 for the buoys*

**Comment 22**

Will use "mast" as suggested

**Comment 23**

Text edited as below in the introduction to address this comment:

*Sentinel-1 A and B are two polar-orbiting satellites equipped with C-band SAR. This sensor which records surface roughness, has the advantage of operating at wavelengths not impeded by cloud cover or a lack of illumination and can acquire data over a site during day or night in all weather conditions. The Sentinel-1 Level 2 OCN product includes a component called Ocean Wind Fields (OWI) which is a ground range gridded estimate of the surface wind speed and direction at 10 m a.s.l, assuming a neutral atmospheric stratification, with a spatial resolution of 1 km2.*

**Comment 24**

Values below 2 m/s can be filtered for the final document.

**Comment 25**

We will use "mast"

**Comment 26**

Here we took an average of the % of error in wind power across the 7 sites. Text changed to:

*The bias and the error on the wind power assessment were increased on average by 9.14% across the 7 sites as shown in Table 7.*

**Comment 27**

Captions updated as below:

**Figure 5: Wind speed histograms of Sentinel-1 SAR Level 2 OCN (right) and in situ (left) data in m s-1 with corresponding Weibull fits for the weather buoy data compared with those produced from the SAR data at the same locations.**

**Figure 6. Wind speed histograms of Sentinel-1 SAR Level 2 OCN (right) and in situ (left) data in m s-1 with corresponding Weibull fits for the coastal weather station data compared with those produced from the SAR data at the same locations.**

**Comment 28 - 29**

Text updated as below:

*The results show that the percentage error on the average wind power was lowest for the coastal weather stations. This may indicate that they could be more reliable than weather buoys, perhaps due to the presence of waves and the relatively low altitude of the buoys. In that case, the error in offshore locations could be overestimated due to inaccuracies with the weather buoy data, although there is no possibility of proving this with certitude. The validation of the Level 2 OCN product should be further investigated in coastal area since land contamination and coastal topography can introduce bias. Another interesting feature is that the bias observed in the match-up comparison seemed to disappear in this climatological analysis. The main difference between the match-up comparison and the analysis performed here arises from including in situ data even when satellite data were not available. In this study, satellite data can be unavailable for two reasons: no data were recorded as a consequence of the relatively low revisit time of the satellite, or the data recorded were discarded if it was flagged as 'bad quality'. The former should not have any effect on the long-term statistics since an increase in sample size will result in a better Weibull distribution. However, the latter might actually introduce an artificial bias in the match-up comparison by limiting it to a specific type of situation in which satellite measurements are easier to perform.*

**Comment 30 - 31**

Text updated as below:

*In this section, the use of the Sentinel-1 Level 2 OCN product to assess wind resources around Ireland at 10 m a.s.l. with a 1 km$^2$ spatial resolution is presented. A clear separation of the mean wind speed into two different areas was clearly visible (Figure 7).*

*In terms of wind power, the results logically revealed a similar pattern with an increased heterogeneity, due to the fact that the wind power is connected to the cube of the wind speed (Figure 8). The northwest area had an average wind power of 700 W m-2 in comparison with 500 W m-2 for the rest of the map, resulting in an overall difference of 20% between the two areas. It is interesting to note that the central area of the Irish sea also has a significant potential in terms of wind power, although lower than that of the northwest area. Regarding coastal areas, a steep horizontal gradient was observed from the shore up to 15-20 km offshore, with the exception of the remote peninsulas on the west coast where the gradient*

*was much shorter or non-existent. In both analyses, the apparent swats can be attributed to the low sample size of satellite data which correlates with Figure 1. The better spatial resolution of SAR data inevitably reduces the revisiting time and therefore the sample size. With time, these artefacts will diminish as the satellite will acquires additional data.*

**Comment 32**

We have removed this comment re: turbulence and directional variance.

**Comment 33**

Comment added to captions 7 to 10 highlighting the visibility of the satellite tracks.

**Comment 34**

Text edited:

*The results also highlighted the necessity for additional in situ validation points for satellite products and showed that there is a need to improve the Sentinel-1 level 2 OCN product algorithm at the edges of the swaths, perhaps through the application of machine learning techniques.*

**Comment 35**

Min, max and number of samples per season will be added to Figure 9

**Comment 36**

We will split this section this into 'Discussion' and 'Conclusions' as suggested.

**Comment 37**

Text edited:

*In any case, it was concluded that the Sentinel-1 Level 2 OCN product can be used to estimate the long-term wind speed distribution and the average wind power. This result could be obtained by using the method of the moments and assuming a Weibull law in order to compensate for the low temporal coverage of the satellites. Even though more investigation is needed to assess the OCN product in coastal area, this study showed that this remotely sensed data can be used to assess the wind resources in coastal areas as close as 1 km to the shore.*

**Comment 38**

Test added:

*Users should exercise caution when working with Sentinel-1 SAR data since a location-dependent error was found at the swath edges. The cause of this discrepancy could not be identified, but perhaps a machine learning technique based on a learning dataset of in situ data could be used to mitigate this effect.*

Added References:

Charnock H (1955) Wind stress on a water surface. Q J R Meteorol Soc 81:639–640

ESA, 2019. *TOPSAR Processing.* [Online] Available at:
https://sentinel.esa.int/web/sentinel/technical-guides/sentinel-1-sar/products-algorithms/level-1-algorithms/topsar-processing
[Accessed 09 12 2019].

---

## Author Response (AR1)

**Response to RC1**

5    Thank you for your comments.

The scientific value of this paper lies in the fact that it provides validation of the Sentinel-1 wind speed data against in-situ measurements. We have also contacted Ifremer and CLS and they have confirmed that they have validated Sentinel-1 data only against numerical models and so no "private" validation
10  against in-situ data exists.

We feel that validation against in-situ measurements is an important step towards the increased use of Sentinel-1 data for wind resource assessment purposes and that such validation results and details of the methodologies used should be in the public domain.

It is encouraging to see that Reviewer 2 agrees with this view,

**Response to RC2**

20  Thank you for your detailed comments. These have now been incorporated into the document. Responses to each comment are provided below.

**Comments 1 to 5**

Text edited as below to address these comments:

*In this study, the Sentinel-1 A and B Level 2 OCN product produced by the European Space Agency (ESA) was validated. This product, derived from SAR observations, provides measurement of neutral surface*
30  *wind speed and direction at 10 m above sea level (a.s.l.) with a spatial resolution of 1 km$^2$. Even though this type of analysis was previously performed in other parts of Europe (Hasager et al., 2015), it has never been conducted using both marine and coastal in situ measurements at a national scale in Ireland, which has a significant offshore wind resource (Remmers et al., 2019). Moreover, to the authors' knowledge, the Sentinel-1 level 2 OCN product has not yet been validated against in situ measurements,*
35  *with the exceptions of one match-up comparison in the waters adjacent to the Korean peninsula (Jang et al., 2019). Similarly, long term statistics retrieved using this product, such as the average wind power, which is the most relevant for the wind energy industry, have never been analysed before.*

*The aim of this study was to validate and the Sentinel-1 A and B Level 2 OCN product against in situ*
40  *measurements in Ireland and asses this data ability to describe the wind resources. First the satellite product and the study area are introduced, next the methodology is provided and finally, the results are presented and discussed.*

**Comment 6 to 9**

Text edited as below to address these comments:

*The two satellites are located on the same orbit 180° apart and at an altitude close to 700 km. In Irish coastal waters, the acquisition mode is Interferometric Wide (IW) swath using the TOPSAR technique which provide a better quality product by enhance the image homogeneity (ESA, 2019). All Sentinel-1A and B SAR images in IW acquisition mode from May 1, 2017 to May 1, 2019, in the area located around Ireland between 51°N and 56°N in latitude and 5°W and 16°W in longitude, were collected (n=5,509). The quality flag for these data ranges from 0 to 3 (0 being the best and 3 the worst) and, following visual inspection, only data with a quality flag ≤ 2 were used for the validation. The Level 2 product tiles were combined into a gridded map for the area of interest, in order to form a data cube where each pixel had a corresponding time series of measurements. The revisit rate ranges from 10 to 20 passes per month for most areas in Irish waters, which occur in the morning around 6.30 am or in the evening around 6 pm, Greenwich Mean Time (GMT) in the winter and Irish Standard Time (IST) in the summer. Figure 1 shows the number of samples for each pixel and Figure 2 shows the average daily passing time of the satellites. The impact on quality flag from landmass contamination was visible with the reduced sample size in coastal area.*

**Comment 10**

Text edited as below to address this comment:

*taken as 0.0002 m (Charnock, 1955). Table 1 gives the exact locations of these buoys and their percentage of availability*

**Comment 11 to 13**

Text edited as below to address these comments:

*The predominant wind direction on the Irish west coast is eastward, flowing from the sea to toward the land. Simulations of these type of flows have shown that for a moderate coastal slope, onshore wind speeds recorded at proximity to the shore can equate the wind speeds at sea just before reaching the coast (Bassi Marinho Pires et al., 2015). Following this principle, the wind speed derived from satellite measurement were not scaled to the weather station terrain elevation, but instead were considered as being in the same streamline and kept at the OCN product elevation of 10 m a.s.l.. The weather station data were compared with Sentinel-1 SAR Level 2 OCN wind speeds measured with the closest pixel without quality flag. Due to the complex Irish coast line and to avoid land contaminate, the OCN measurement were one or two pixels away from the shore (i.e. 1 or 2 km).*

**Comment 14**

We feel that the donation used is acceptable as it is defined in the text.

85

**Comment 15**

 We would like to keep these definitions in the manuscript in order to save the reader having to look them up.

90

**Comment** 16

We have added this text to the introduction as suggested but would also like to keep it here as it is a nice transition to the next section.

95

**Comment 17 – 21**

Text edited as below to address these comments:

*3.1 Match-up comparison*
*The main objective of the Sentinel-1 SAR surface wind comparison with in situ data was to highlight the*
100 *agreement and dissonance between the two. Sentinel-1 SAR Level 2 OCN surface wind data and in situ wind data were collocated in space and time. Since the spatial resolution of this product is very high (1 km$^2$) and offshore winds have a low spatial heterogeneity caused by sea surface homogeneity, the spatial resolution was slightly degraded in order to increase the number of samples. The best remotely sensed value, both in term of quality and distance, from the pixel directly adjacent to the in situ measurement*
105 *(i.e. 3 km$^2$) was chosen for the match-up comparison.*

*In the time domain, each in situ measurement with the corresponding satellite measurement performed in a 30 mn time interval before or after were selected for the analysis. For all buoys, the wind speed correlation with the remotely sensed data at a one-hour time interval was around 0.99, which showed*
110 *that the time difference between the satellite and in situ data does not introduce a significant source of error. Another factor in this respect is that Sentinel-1 SAR Level 2 OCN spatial averaging at the resolution of 1 km$^2$ may somewhat compensate for the lack of time averaging. However, the bias due to these differences in the measurement technique, in space and time, is difficult to predict theoretically. Therefore, the bias can be caused not only by the SAR sensor intrinsic error, but also by the different*
115 *scales of measurement. Another source of potential error derived from the assumption of neutral atmospheric stability when scaling the buoy data from 3 m to 10 m a.s.l using Equation (1). Hence, the overall bias needed to be evaluated empirically through a match-up comparison.*

*The bias for all available data used in the match-up comparison was found to be -0.42 m s-1 and -0.39 m*
120 *s-1 and the RMSE 1.41 m s-1 and 1.51 m s-1 for the buoys*

**Comment 22**

Will use "mast" as suggested

**Comment 23**

Text edited as below in the introduction to address this comment:

*Sentinel-1 A and B are two polar-orbiting satellites equipped with C-band SAR. This sensor which records surface roughness, has the advantage of operating at wavelengths not impeded by cloud cover or a lack of illumination and can acquire data over a site during day or night in all weather conditions. The Sentinel-1 Level 2 OCN product includes a component called Ocean Wind Fields (OWI) which is a ground range gridded estimate of the surface wind speed and direction at 10 m a.s.l, assuming a neutral atmospheric stratification, with a spatial resolution of 1 km2.*

**Comment 24**

Values below 2 m/s can be filtered for the final document.

**Comment 25**

We will use "mast"

**Comment 26**

Here we took an average of the % of error in wind power across the 7 sites. Text changed to:

*The bias and the error on the wind power assessment were increased on average by 9.14% across the 7 sites as shown in Table 7.*

**Comment 27**

Captions updated as below:

**Figure 5: Wind speed histograms of Sentinel-1 SAR Level 2 OCN (right) and in situ (left) data in m s-1 with corresponding Weibull fits for the weather buoy data compared with those produced from the SAR data at the same locations.**

**Figure 6. Wind speed histograms of Sentinel-1 SAR Level 2 OCN (right) and in situ (left) data in m s-1 with corresponding Weibull fits for the coastal weather station data compared with those produced from the SAR data at the same locations.**

**Comment 28 - 29**

Text updated as below:

*The results show that the percentage error on the average wind power was lowest for the coastal weather stations. This may indicate that they could be more reliable than weather buoys, perhaps due to the presence of waves and the relatively low altitude of the buoys. In that case, the error in offshore locations could be overestimated due to inaccuracies with the weather buoy data, although there is no possibility of proving this with certitude. The validation of the Level 2 OCN product should be further investigated in coastal area since land contamination and coastal topography can introduce bias. Another interesting feature is that the bias observed in the match-up comparison seemed to disappear in this climatological analysis. The main difference between the match-up comparison and the analysis performed here arises from including in situ data even when satellite data were not available. In this study, satellite data can be unavailable for two reasons: no data were recorded as a consequence of the relatively low revisit time of the satellite, or the data recorded were discarded if it was flagged as 'bad quality'. The former should not have any effect on the long-term statistics since an increase in sample size will result in a better Weibull distribution. However, the latter might actually introduce an artificial bias in the match-up comparison by limiting it to a specific type of situation in which satellite measurements are easier to perform.*

**Comment 30 - 31**

Text updated as below:

*In this section, the use of the Sentinel-1 Level 2 OCN product to assess wind resources around Ireland at 10 m a.s.l. with a 1 $km^2$ spatial resolution is presented. A clear separation of the mean wind speed into two different areas was clearly visible (Figure 7).*

*In terms of wind power, the results logically revealed a similar pattern with an increased heterogeneity, due to the fact that the wind power is connected to the cube of the wind speed (Figure 8). The northwest area had an average wind power of 700 W m-2 in comparison with 500 W m-2 for the rest of the map, resulting in an overall difference of 20% between the two areas. It is interesting to note that the central area of the Irish sea also has a significant potential in terms of wind power, although lower than that of the northwest area. Regarding coastal areas, a steep horizontal gradient was observed from the shore up*

*to 15-20 km offshore, with the exception of the remote peninsulas on the west coast where the gradient was much shorter or non-existent. In both analyses, the apparent swats can be attributed to the low sample size of satellite data which correlates with Figure 1. The better spatial resolution of SAR data inevitably reduces the revisiting time and therefore the sample size. With time, these artefacts will diminish as the satellite will acquires additional data.*

**Comment 32**

We have removed this comment re: turbulence and directional variance.

**Comment 33**

Comment added to captions 7 to 10 highlighting the visibility of the satellite tracks.

**Comment 34**

Text edited:

*The results also highlighted the necessity for additional in situ validation points for satellite products and showed that there is a need to improve the Sentinel-1 level 2 OCN product algorithm at the edges of the swaths, perhaps through the application of machine learning techniques.*

**Comment 35**

Min, max and number of samples per season can be added to Figure 9

**Comment 36**

We will split this section this into 'Discussion' and 'Conclusions' as suggested.

**Comment 37**

Text edited:

[revised manuscript text omitted]

---

## Referee Report (RR1)

[referee-annotated manuscript omitted]

---

## Author Response (AR2)

**Response to RC1**

Thank you for your comments.

The scientific value of this paper lies in the fact that it provides validation of the Sentinel-1 wind speed data against in-situ measurements. We have also contacted Ifremer and CLS and they have confirmed that they have validated Sentinel-1 data only against numerical models and so no "private" validation against in-situ data exists.

We feel that validation against in-situ measurements is an important step towards the increased use of Sentinel-1 data for wind resource assessment purposes and that such validation results and details of the methodologies used should be in the public domain.

It is encouraging to see that Reviewer 2 agrees with this view.

**Response to RC2**

Thank you for your detailed comments. These have now been incorporated into the document with changes highlighted below. For ease of review, responses to each comment have also been provided in a marked-up version of your original supplemental document which has also been uploaded.

[revised manuscript text omitted]

---

## Author Response (AR3)

I reviewed your new manuscript submission and your response to the reviewer comments. I find you have answered satisfactorily to most of the reviewer's concerns. But there are a couple of small sections where I would like to see some changes:

1. In L96, you write "the OCN measurement were one or two pixels away from the shore". I agree with the reviewer in this respect. I suggest that you write your argument regarding the averaging nature of the SAR measurements "As the product is already an average of SAR measurements..."

Text altered to: "As the Level 2 OCN product values are already an average of SAR measurements (resolution 10m and product resolution 1km) further averaging was not applied."

2. In L268-270, you write "... showed that there is a need to improve the algorithms used by the Sentinel-1 level 2 OCN product to process the raw SAR data, particularly at the edges of the swaths." Again, I do agree with the reviewer's comments. "The edge effects are inherent from the raw SAR observations, and it would therefore not help to modify the wind retrieval algorithms." Therefore, I find a more appropriate conclusion is that the user should be aware of the edge limitations of the data and filter the data accordingly. Please rephrase in the updated manuscript.

Text altered to: "It can be concluded that the accuracy was dependent upon location, which is a factor that should be considered when using Santinel-1 SAR data, this is shown to be particularly the case at the edge of swaths, users should be aware of this limitation and filter the data accordingly."

I would also request that you make some corrections to the manuscript following the WES guidelines (https://www.wind-energy-science.net/for_authors/manuscript_preparation.html):
1. Please review the guidelines regarding standard units. There are inconsistencies throughout the manuscript. The figures and figure captions should also follow the instructions.

Standard units used throughout. Can more detail be provided with regards the issues with captions?

2. The labelling of the various panels in the figures and figure captions does not follow the WES standard. "Labels of panels must be included with brackets around letters being lowercase (e.g. (a), (b))."

We have replace "left" and "right" with (a) and (b) where applicable.

3. Some references do not follow the WES standard, in particular the journal name abbreviations. Also, to facilitate the work of the reviewers, all the references must contain clickable DOIs or URLs when possible.

Abbreviations adjusted and clickable URLs included.